# Functional Characterisation of Three Glycine *N*-Acyltransferase Variants and the Effect on Glycine Conjugation to Benzoyl–CoA

**DOI:** 10.3390/ijms22063129

**Published:** 2021-03-18

**Authors:** Johann M. Rohwer, Chantelle Schutte, Rencia van der Sluis

**Affiliations:** 1Laboratory for Molecular Systems Biology, Department of Biochemistry, Stellenbosch University, Private Bag X1, Matieland, Stellenbosch 7602, South Africa; jr@sun.ac.za; 2Focus Area for Human Metabolomics, North-West University, Private Bag X6001, Potchefstroom 2520, South Africa; schutte.tella@gmail.com

**Keywords:** glycine conjugation, glycine *N*-acyltransferase (GLYAT), benzoate, hippurate, coenzyme A

## Abstract

The glycine conjugation pathway in humans is involved in the metabolism of natural substrates and the detoxification of xenobiotics. The interactions between the various substrates in this pathway and their competition for the pathway enzymes are currently unknown. The pathway consists of a mitochondrial xenobiotic/medium-chain fatty acid: coenzyme A (CoA) ligase (ACSM2B) and glycine *N*-acyltransferase (GLYAT). The catalytic mechanism and substrate specificity of both of these enzymes have not been thoroughly characterised. In this study, the level of evolutionary conservation of GLYAT missense variants and haplotypes were analysed. From these data, haplotype variants were selected (156Asn > Ser, [17Ser > Thr,156Asn > Ser] and [156Asn > Ser,199Arg > Cys]) in order to characterise the kinetic mechanism of the enzyme over a wide range of substrate concentrations. The 156Asn > Ser haplotype has the highest frequency and the highest relative enzyme activity in all populations studied, and hence was used as the reference in this study. Cooperative substrate binding was observed, and the kinetic data were fitted to a two-substrate Hill equation. The coding region of the *GLYAT* gene was found to be highly conserved and the rare 156Asn > Ser,199Arg > Cys variant negatively affected the relative enzyme activity. Even though the 156Asn > Ser,199Arg > Cys variant had a higher affinity for benzoyl-CoA (*s*_0.5,benz_ = 61.2 µM), *k*_cat_ was reduced to 9.8% of the most abundant haplotype 156Asn > Ser (*s*_0.5,benz_ = 96.6 µM), while the activity of 17Ser > Thr,156Asn > Ser (*s*_0.5,benz_ = 118 µM) was 73% of 156Asn > Ser. The in vitro kinetic analyses of the effect of the 156Asn > Ser,199Arg > Cys variant on human GLYAT enzyme activity indicated that individuals with this haplotype might have a decreased ability to metabolise benzoate when compared to individuals with the 156Asn > Ser variant. Furthermore, the accumulation of acyl-CoA intermediates can inhibit ACSM2B leading to a reduction in mitochondrial energy production.

## 1. Introduction

The glycine conjugation pathway is a two-step enzymatic reaction responsible for the metabolism/detoxification of natural substrates from (i) food (for example., salicylate [1], dietary polyphenols and medium-chain fatty acids (MCFAs)), (ii) xenobiotics (for example benzoate), and (iii) metabolites produced from organic acidaemia [2,3,4] (Figure 1). Benzoate and salicylate are activated to an acyl-coenzyme A (CoA) by the mitochondrial xenobiotic/medium-chain fatty acid: CoA ligases (ACSM2B,EC 6.2.1.2) [5,6] and subsequently conjugated to glycine by glycine *N*-acyltransferase (GLYAT, EC 2.3.1.13) to form hippuric acid and salicyluric acid, respectively [3,7,8]. Gut microorganisms produce benzoyl–CoA, which is a substrate for glycine conjugation, from dietary polyphenols [9]. MCFAs, for example, caprylic acid, are activated by ACSM2B ligase in the liver before entering the mitochondrial beta-oxidation cycle [10]. The overall rate of glycine conjugation can be influenced by the availability of cofactors, variations in the *ACSM2B* and *GLYAT* genes, and a difference in expression levels of ACSM2B and GLYAT [11,12,13,14].

All individuals will need to metabolise benzoate and salicylate from natural sources (for example, benzoate and salicylate present in berries and milk products [1]), and depending on whether an individual’s diet is high in polyphenols, benzoate will also be formed by the gut microbes [15]. However, the exposure of humans to these compounds is increasing [16,17] since benzoate is widely used as a preservative in food and pharmaceuticals [18]. Benzoic acid consumption has been linked to adverse effects such as diarrhoea, metabolic acidosis, tremors, and childhood hyperactivity syndrome [19,20]. These symptoms might be the result of low levels of glycine as glycine conjugation is needed to detoxify the high levels of benzoate. The glycine shortage results in a reduction in creatinine, glutamine, urea, and uric acid production (Figure 1) [1]. A shortage of glycine can, furthermore, increase the accumulation of acyl-CoA intermediates. Accumulation of xenobiotic-CoA esters results in the sequestration of CoA and the inhibition of the acid: CoA ligases [21]. If the ACSM2B ligase is inhibited by the acyl-CoA intermediates, ACSM2B can no longer activate MCFAs for use during beta-oxidation. The glycine conjugation pathway, therefore, plays a fundamental role in the homeostatic energy balance within the mitochondria by preventing coenzyme A (CoASH) sequestration. Although hippuric acid and benzoic acid have a similar water solubility value, glycine conjugation decreases the toxicity of benzoate by forming less lipophilic conjugates that can be more readily transported out of the mitochondria [22]. Studies have shown that benzoic acid inhibits the elimination of salicyluric acid, and that the accumulation of salicyl-CoA may result in toxicity and liver damage [23,24].

A previous study, in a cohort of isovaleric acidaemia patients from South Africa, evaluated the effectiveness of glycine supplementation on the clinical outcome of these patients. Even though these patients were all homozygous for the same isovaleryl-CoA dehydrogenase mutation, variation in the responsiveness to glycine supplementation was observed [25]. Glycine supplementation is used as a treatment in these patients in order to conjugate the toxic accumulated isovaleryl-CoA to glycine by GLYAT forming isovalerylglycine [26] (Figure 1). It is hypothesised that missense variants in the *GLYAT* gene do not contribute significantly to the inter-individual glycine conjugation rates observed in these patients as the *GLYAT* gene was shown to be highly conserved in a small cohort of 1537 individuals [27]. Variation in the expression level of GLYAT may influence the glycine conjugation rate in humans, as was shown for rats, where diet affected the expression of GLYAT in the liver [28,29]. Furthermore, benzoyl–CoA is the preferred substrate for GLYAT, followed by salicyl–CoA and then isovaleryl–CoA [30]. Therefore, a diet high in benzoate can be one of the factors that better explain the inter-individual variation in responsiveness to glycine supplementation in these patients as the derived benzoyl–CoA will outcompete isovaleryl–CoA as a substrate. Currently, no studies are available that provide data on the interaction/competition of the various substrates involved in the glycine conjugation pathway.

It is extremely difficult to obtain fresh human liver samples in order to study the catalytic mechanism of GLYAT using a purified enzyme [14]. Previously determined GLYAT kinetic parameters, using either mitochondrial lysate preparations from liver tissue or purified recombinantly expressed GLYAT enzyme, vary considerably [28,30,31,32,33,34], with only three studies determining the *K*_m_ value for glycine [28,32,33]. These studies also assumed that GLYAT followed a Michaelis–Menten reaction mechanism and, as a consequence, reported a sequential two-substrate mechanism. A preliminary study, in which the recombinantly expressed purified 156Asn > Ser GLYAT variant was characterised, indicated that GLYAT exhibits mechanistic kinetic cooperativity and not a Michaelis–Menten reaction mechanism [35]. Most of the studies also did not report on whether the wildtype GLYAT or a variant was used in the analyses. This is important in order to compare the different studies as it has been shown that single nucleotide polymorphisms (SNPs) can alter the kinetic parameters of GLYAT [31,34].

The aim of this study was to analyse the genetic diversity and haplotype variation of the *GLYAT* gene using a larger population (125,748 exomes and 15,708 whole genomes) to determine the level of conservation in the worldwide population. Haplotypes occurring at low frequencies in a population are more likely to be deleterious and therefore associated with adverse detoxification. This analysis also allowed us to identify the haplotypes for further characterisation in terms of enzyme activity and mechanism. In an effort to address the limitations of previous studies, the bi-substrate (glycine and benzoyl–CoA) reaction kinetics of the purified 156Asn > Ser, 17Ser > Thr,156Asn > Ser, and 156Asn > Ser,199Arg > Cys GLYAT variants were determined over a wide range of substrate concentrations in order to determine the effect of genetic variants on the relative enzyme activity and kinetic parameters.

## 2. Results and Discussion

Understanding and quantifying the interaction and competition between various substrates or xenobiotics for detoxification by glycine conjugation requires information on differences in the enzyme activity and catalytic mechanism of GLYAT variants. Through the analyses of the variant data that are available on the gnomAD browser and Ensembl database, the allele frequencies in the worldwide population and the level of conservation of the GLYAT haplotypes could be determined. Identifying the haplotype frequencies also resulted in the enzymatic characterisation of relevant variants of the enzyme. Based on the determined haplotype frequencies, three haplotypes were chosen and characterised in terms of relative enzyme activity and catalytic mechanism in order to determine the effect of the missense variants on the enzyme activity.

### 2.1. Level of Conservation of the GLYAT Gene

The exome and whole genome data available on the gnomAD browser (gnomad.broadinstitute.org/; accessed on 10 September 2020) [36] and the haplotype data available on the Ensembl database (ensemble.org; accessed on 10 September 2020) [37] were analysed in order to determine the allelic variation and haplotype diversity in the worldwide population. The gnomAD browser is the largest database that includes allele frequencies of variants located in protein-coding regions, as both exome and genome sequencing data from a wide variety of large-scale sequencing projects are combined. The allele frequencies of the missense variants found in GLYAT (ENST00000344743.3) were downloaded from the gnomAD browser [36] and analysed (Appendix A).

For *GLYAT,* 193 missense variants were identified. Of these, only two variants had an allele frequency > 0.5%, that is., Asn156Ser (94.87%) and Ser17Thr (19.67%), while the remaining variants were rare with allele frequencies ranging between 0.2% and 0.0004% (a rare variant is only found in one allele of one individual out of a total of 141,456 analysed). The Asn156Ser and Ser17Thr variants had the highest allele frequencies in all of the populations analysed (Table 1 and Figure 2). The same trend was observed in a smaller previous study that included data from the 1000 genomes and HapMap projects, in addition to data from 61 Caucasian Afrikaners, 4 Khoi-San, and 1 Bantu individual from South Africa [27]. The Asn156Ser missense variant also had the highest homozygous genotype frequency of 90.3%, followed by Ser17Thr (4.5%). Only six other variants were found as homozygotes, namely Arg131His (East Asian—0.005%), Arg131Cys (South Asian—0.002%), Met65Thr (Latino—0.001%), Thr73Ile (East Asian—0.001%), His101Tyr (South Asian—0.002%) and Thr244Met (Latino—0.001%).

In the African/African American population, the Asn156Ser allele occurred at a frequency of 99.58%, and the Ser17Thr allele at 19.17%, with the rest of the variants occurring at a frequency below 0.07%. Several studies have shown that in African populations, the genetic diversity and discovery rate of novel variants is higher [38,39,40,41]. However, the expected high level of genetic diversity was not observed in the *GLYAT* gene, indicating that the gene is conserved even in genetically diverse individuals. In contrast, a lower allele frequency for Asn156Ser (77.16%) and slightly higher frequency for Ser17Thr (26.83%) and Arg131His (2.62%) were found in the East Asian population. Diet and the environment can be a strong driver of selection [42]. An example is the high prevalence of the slow acetylation phenotype in populations practicing farming and herding [43]. In other species, such as domestic cats, a reduced ability to metabolise several drugs and structurally related phenolic compounds has been observed [44,45,46,47] due to gene inactivation as a consequence of minimal exposure to plant-derived toxicants. It was also shown in rats that diet influences the expression level of GLYAT in the liver [28,29]. Whether diet played a role in the selection of the variants found in the East Asian population or if diet might affect the expression level of GLYAT in the liver of humans needs to be further investigated.

The 1000 Genomes haplotype data [37,48] for *GLYAT* are summarised in Appendix A. The haplotype frequencies were analysed across the 26 populations to find the haplotypes with the highest frequency. In total, 25 haplotypes were reported of which four have a frequency >0.5% [156Asn > Ser (69.9%); 17Ser > Thr,156Asn > Ser (21.5%); REF (7.15%); 131Arg > His,156Asn > Ser (0.52%)]. This study and previous studies [27,49] clearly show that the 156Asn > Ser variant should be regarded as the reference sequence due to the high allele frequency identified in all populations studied. The rare haplotype frequencies ranged from 0.22 to 0.02%. The 156Asn > Ser haplotype had the highest frequency in all the populations, but notably, the frequency was lower in the East Asian (51.6%) and the South Asian (65.6%) populations when compared to the African (79.5%) and European (80.8%) populations. Of the 25 haplotypes, 17 were predicted by the sorting intolerant from tolerant (SIFT) [50] and polymorphism phenotyping (PolyPhen) [51] algorithm tools, to have a deleterious effect on protein function. Only two rare haplotypes [73Thr > Ile (0.18%) and 17Ser > Thr (0.02%)] were not found in combination with the 156Asn > Ser variant. 

The Tajima’s D value was calculated using MEGA X [52] to determine whether the human *GLYAT* gene is evolving neutrally (Table 2). The highly negative Tajima’s D value of –2.13 indicates that variants located within the *GLYAT* gene are under negative selection. This is supported by the large number of low-frequency alleles observed in all of the populations analysed in this study.

Comparative phylogenetic studies between apes and humans clarify the patterns of evolutionary change in the human lineage [53,54,55]. To construct the phylogenetic tree, the *GLYAT* haplotype sequence data were used to perform maximum likelihood fits to determine the best amino acid substitution model to use [56]. The model predicted to have the best fit, was the Jones–Thornton–Taylor (JTT) model [57] with discrete gamma rate categories (+G). In order to determine the evolution of the *GLYAT* gene since the human and chimpanzee split, phylogenetic analyses were subsequently performed (Figure 3). The robustness of the tree was assessed using 500 bootstrap replicates. For *GLYAT*, the human/chimpanzee/gorilla/bonobo/orangutan clade had good bootstrap support of 100%, while ancestral nodes within this clade were poorly supported with values ranging from 23–63% (Figure 3) [58]. These ancestral nodes were made up of human haplotypes with very low haplotype frequencies (<0.6%). The phylogenetic analyses further indicate that the *GLYAT* gene is conserved across all population groups. The *Pan troglodytes, Pan paniscus, Pongo abelii,* and *Gorilla gorilla* haplotypes are found within the human clade which indicates that relatively few changes have occurred within the *GLYAT* gene since the chimpanzee and human split.

The allelic and haplotype diversity analyses, together with the negative Tajima’s D value, all point to the fact that a large number of rare variants/haplotypes are found in the worldwide population. The phylogenetic analyses (Figure 3) suggested that the coding regions of the *GLYAT* gene are well conserved through evolution. Therefore, even in a large diverse population of 141,456 individuals, the *GLYAT* gene was shown to be highly conserved. This confirmed the results of the previous smaller study where only 1537 individuals were included [27].

### 2.2. Relative Enzyme Activity and Catalytic Parameters

We selected two of the four haplotypes with a frequency > 0.5% [156Asn > Ser (69.9%); 17Ser > Thr,156Asn > Ser (21.5%)] and one rare haplotype [156Asn > Ser,199Arg > Cys (0.02%)] for further kinetic analysis. The recombinantly expressed and purified 17Ser > Thr and 199Arg > Cys variants were previously characterised [31]. It was shown that the 17Ser > Thr variant had activity comparable to that of the wild-type enzyme (listed as “REF” in Appendix A) and the 199Arg > Cys mutation had less than 5% activity of that of the wild-type enzyme. The 17Ser > Thr,156Asn > Ser and 156Asn > Ser,199Arg > Cys haplotypes were both predicted by the SIFT [50] and PolyPhen [51] algorithm tools to have a deleterious effect on protein function (Appendix A).

To determine if there are indeed differences in catalytic activity between the GLYAT haplotype variants, we initially compared the relative enzyme activities of the three haplotypes at one specific set of substrate concentrations (Figure 4). There were significant differences between the three haplotypes, with 156Asn > Ser,199Arg > Cys showing only 12.3% of the activity of 156Asn > Ser, and 17Ser > Thr,156Asn > Ser exhibiting an intermediate level of activity (49.4%). When comparing the enzyme activity of the 199Arg > Cys mutation on its own [31] with that of the 156Asn > Ser,199Arg > Cys haplotype, an increase in enzyme activity of 5% was observed. This indicates how important it is to characterise haplotypes rather than SNPs, especially in the field of pharmacokinetics and pharmacogenomics. The most recent study [34] which compared the enzyme activity of the 61Gln > Leu variant with that of the wild-type, found that the 61Gln > Leu variant showed a decrease in specific activity when compared to the wild-type. It is important to note that the 61Gln > Leu mutation has only been identified in two haplotypes in the South African Afrikaner population that is, 61Gln > Leu,156Asn > Ser and 17Ser > Thr, 61Gln > Leu,156Asn > Ser [27], and therefore, the activity might be affected by the other SNPs in these haplotypes. The crystal structure of GLYAT is not available but based on a molecular model, the 199Arg > Cys mutation alters a highly conserved Arg in an α-loop- α motif which is important for substrate binding in the Gcn5-related *N*-acetyltransferases (GNAT) superfamily [60,61]. Moreover, 156Asn > Ser is on a poorly predicted loop from Lys159 to Met167, making interpretation of its role in enzyme activity difficult. The relative enzyme activity of the haplotypes also correlated with the PolyPhen and SIFT predictions, in addition to the haplotype frequency, with rarer haplotypes being less active. 

Because we observed such striking differences between the catalytic activities of the three GLYAT variants, the kinetic parameters of each of the haplotypes were characterised in greater detail to determine possible differences in the catalytic mechanism. Initial rates were determined for a range of concentrations of both substrates. Cooperative substrate binding was observed. To account for this, the data were fitted to a two-substrate Hill equation (see Section 3). Since the concentration of one substrate in two-substrate kinetic experiments will affect the values obtained for the kinetic parameters of the other substrate, we varied both substrate concentrations in a grid and performed a global fit using non-linear regression to obtain a single set of enzyme-kinetic parameters that best describe all the data.

The final fits for the three haplotypes were visualised with 3D-surface plots (Figure 5), showing the fit of the kinetic model to all of the experimental data. Data were processed and fitted as described in Section 3. Each data point represents the mean of triplicate measurements. The best model fit is indicated by the coloured surface. The cooperative kinetics can be clearly observed, especially for benzoyl–CoA as substrate. Because of the difficulty in visualising these data in three dimensions, especially the agreement between model and data, two-dimensional activity plots are included as well (Appendix A), showing rate against glycine concentration at each separate benzoyl–CoA concentration, and vice versa. Importantly, while each of these plots shows individual experimental data points and a model fit, it should be noted that the line indicates the global fit of the model to all the data, not only the data for a particular plot, which explains the discrepancies observed in some cases. For further illustration, Lineweaver–Burk plots of selected datasets are included as Appendix A; the cooperative response can be observed as a deviation from linearity in the fit. The kinetic parameters obtained from the fitting are summarised in Table 3. Parameters and standard errors were estimated from a global fit of initial rate data to the two-substrate Hill equation. Refer to Section 3 for details and definition of the kinetic parameters.

A number of features of the kinetic dataset in Table 3 merit comment. First, the catalytic activity (*k*_cat_) mirrored the trend observed in the initial activity study (Figure 4), with the haplotype variant 156Asn > Ser,199Arg > Cys displaying only 10% of the activity of 156Asn > Ser, while that of 17Ser > Thr,156Asn > Ser was 73% of 156Asn > Ser. The *k*_cat_ was also by far the most affected parameter by the sequence changes introduced in the haplotypes. 

Secondly, the binding affinity was affected to a lesser extent, being slightly weaker with the half-saturation constant for glycine (*s*_0.5,gly_) and increasing by 28% and 33% for 17Ser > Thr,156Asn > Ser and 156Asn > Ser,199Arg > Cys, respectively, when compared to the most abundant haplotype 156Asn > Ser. In contrast, the effects on the half-saturation constant for benzoyl–CoA (*s*_0.5,benz_) were mixed, with 17Ser > Thr,156Asn>Ser showing an increase of 22%, while 156Asn > Ser,199Arg > Cys showed a decrease of 37% in this parameter. Overall, the binding affinities for both substrates were in the same range for all three GLYAT variants.

Thirdly, both substrates exhibited cooperative binding for all of the haplotypes studied. The cooperativity was stronger for benzoyl–CoA (with Hill coefficients ranging between 1.5 and 3.5) than for glycine (with Hill coefficients ranging between 1.3 and 1.6). Moreover, all three haplotype variants showed similar cooperativity for glycine, while there were marked differences for benzoyl–CoA. In comparison to the 156Asn > Ser variant, 17Ser > Thr,156Asn > Ser showed decreased cooperativity, while 156Asn > Ser,199Arg > Cys showed markedly increased cooperativity.

Since all the known mammalian orthologs of GLYAT, including human (studied here), bovine, and chimpanzee, are monomeric enzymes [7,30,32], the cooperative kinetic responses observed merit further discussion. Ferdinand [62] already showed more than 50 years ago that bi-substrate monomeric enzymes can under certain circumstances exhibit sigmoidal kinetics. This so-called kinetic cooperativity can specifically occur if the two pathways in which, say, substrate A or substrate B binds to the enzyme first, both do occur, but the enzyme shows a preference for one of these pathways. More recently [63], kinetic cooperativity in human glucokinase has been shown to result from the unliganded enzyme existing in two states—a ground state and an activated state. If the rate of interconversion between these states is of the same order as the catalytic rate constant, sigmoidal kinetic responses towards glucose concentration can be observed. More generally, cooperative kinetic responses may also be the result of ‘allokairy’ [64], that is, information changes being transmitted through time, such as an enzyme in an active state after turnover relaxing back to an inactive state over time. The exact mechanism by which GLYAT exhibits sigmoidal kinetics still needs to be elucidated.

Taking all these data into account, the results presented in Table 3 suggest that changes in activity between the three haplotype variants are predominantly due to *V*-effects (that is changes in *k*_cat_), with substrate binding playing a lesser role. The only exception to this might be the binding of benzoyl–CoA to the 156Asn > Ser,199Arg > Cys variant, in which a decreased *s*_0.5_ and significantly increased *h*-value suggest a more potent response as compared to 156Asn > Ser. However, with the tenfold lower *k*_cat_, this increased potency is unlikely to offer significant advantages due to an increase in the population of non-productive binding conformations [65]. Allosteric regulation of certain enzymes is an evolutionary mechanism of adaptation for the selection of specific substrates because the enzyme specificity for substrates controls metabolic flow by sorting metabolites into distinct paths [66]. The glycine conjugation pathway maintains a delicate balance in CoA levels within the mitochondria [21] and this might explain why deleterious variants such as the 156Asn > Ser,199Arg > Cys haplotype are maintained at very low frequencies in the population.

### 2.3. Comparison of GLYAT Kinetic Parameters to Literature Values 

Human GLYAT has been kinetically characterised in previous studies, both in our laboratory and by other investigators. We, therefore, compared the kinetic parameters obtained in this study to previous values from the literature (Table 4). In terms of substrate affinity, previous studies reported apparent Michaelis constants and did not investigate cooperative effects, which makes a direct comparison with our values difficult. Nevertheless, we list the half-saturation constants obtained in this study together with the other *K*_Mapp_-values, reasoning that at least a semi-quantitative comparison is justified on the grounds that firstly the reported Michaelis constants are apparent (and not true) values, and secondly, both *K*_Mapp_ and *s*_0.5_ are operationally defined as the substrate concentration giving a rate of half *V*_max_ at saturating levels of the second substrate.

Overall, both the maximal activity and the affinity parameters agreed well with published literature, with our reported values falling in the ranges reported in previous studies (Table 4). It should be noted that two studies [30,34] reported in Table 4 might contain calculation errors as explained in the footnote. Importantly, though, the differences in the kinetic mechanism for the different haplotypes (Table 3) have not been reported previously. 

Previous substrate specificity studies performed for GLYAT [30,33] need to be repeated in the light of the fact that the measure for substrate specificity of non-cooperative enzymes is *k*_cat_/*K*_m_ or its apparent value if other co-substrates are present [67], but for cooperative enzymes, the appropriate measure is *k*_cat_/*s^h^_0.5_* [68]. This is due to the fact that the degree of cooperativity affects the order of the specificity of the substrates. This effect needs to be taken into account especially when evaluating the substrate specificity of competing substrates in physiological conditions. For example, in the case of isovaleric acidaemia, it would be important to know whether benzoyl–CoA will outcompete isovaleryl–CoA as substrate and whether this has an effect on the observed differences in the effectiveness of using glycine supplementation as therapy in these patients [25]. 

It is very difficult to extrapolate the in vitro values to the in vivo environment where several substrates and pathways need to be considered. Previous studies have found that if the amount of benzoate administered to individuals is increased, hippuric acid excretion will also increase to a maximum after which the excretion level will remain constant. The administration of glycine, on the other hand, resulted in a rapid increase in the hourly excretion rate of hippuric acid [69,70,71]. These studies are very old and the GLYAT haplotypes of the individuals analysed in these studies were not known at the time. Even though significant inter-individual variability in glycine conjugation capacity has been demonstrated after administering benzoate [72], this is probably due to the fact that it is very difficult to control for individual differences in diet during these studies. The metabolites of dietary polyphenols produced by microorganisms contribute the largest portion of the natural substrates that are metabolised by the glycine conjugation pathway [9]. Aspirin and benzoate have been used to characterise individual glycine conjugation capacity; however, adverse reactions, aspirin intolerance, and Reye’s syndrome in children are substantial drawbacks. Therefore, the use of p-aminobenzoic acid (PABA) as an alternative glycine conjugation probe was investigated in a previous study. For the study, 10 human volunteers participated in a PABA challenge test, and p-aminohippuric acid (PAHA), p-acetamidobenzoic acid, and p-acetamidohippuric acid were quantified in urine samples. The *GLYAT* gene of the volunteers was also screened for two polymorphisms associated with normal (17Ser > Thr) and increased (156Asn > Ser) enzyme activity. Although all of the individuals were homozygous for the SNP that results in increased enzyme activity in vitro (156Asn > Ser), excretion of PAHA varied significantly (16–56%, hippurate ratio). The intricacies of PABA metabolism revealed possible limiting factors for the use of this probe substance for the targeted profiling of glycine conjugation [73]. A method to accurately quantify the benzoate detoxification ability of humans in vivo will aid greatly in the understanding of this pathway.

## 3. Materials and Methods

The reference transcript of *GLYAT* (NM_201648.3; ENST00000344743) was used to determine the allelic variation and haplotype frequencies.

### 3.1. Missense Variants Identified in GLYAT Using gnomAD

Exome and whole-genome sequencing data in the genome aggregation database v2 (gnomAD browser—gnomad.broadinstitute.org/; accessed on 10 September 2020) [36] were used to analyse the allele frequencies of the missense variants found in the GLYAT gene. The gnomAD v2 data set contains data from 125,748 exomes and 15,708 whole genomes, all mapped to the GRCh37/hg19 reference sequence. Only high quality genotypes were included in this dataset (GQ ≥ 20, DP ≥ 10, allele balance > 0.2 for heterozygote genotypes).

### 3.2. Haplotype Data Obtained from Ensembl

The 1000 Genomes [48] haplotype data for GLYAT were downloaded from the Ensembl GRCh38 Homo sapiens assembly [74]. The 1000 genomes dataset contains data for 2504 individuals from 26 populations.

### 3.3. Tajima’s Test of Neutrality

To determine if the GLYAT gene is evolving randomly or under directional selection, the Tajima’s *D* test was determined [75,76,77] using MEGA X [52].

### 3.4. Phylogenetic Analyses

The amino acid sequences for each GLYAT haplotype, in addition to a selection of orthologs (*Pan troglodytes, Pan paniscus, Pongo abelii, Gorilla gorilla gorilla, Mus musculus, Rattus norvegicus, Bos taurus, Tursiops truncates, Loxodonta africana, Pteropus vampyrus*, and *Danio rerio*), were downloaded from Ensembl [37]. The amino acid sequences were aligned using ClustalX v2.1 [78]. The amino acid substitution model [56] and the phylogenetic analyses were performed using MEGA X [52]. The robustness of the tree topology was evaluated using bootstrap analysis with a resampling size of 500 replicates.

### 3.5. Expression and Nickel-Affinity Purification of the Recombinant GLYAT Haplotypes

The 156Asn > Ser; 17Ser > Thr,156Asn > Ser and 156Asn > Ser,199Arg > Cys GLYAT recombinant proteins were expressed and purified, as previously reported [31], with 0.5% glycine added to the expression medium in this study. The purified proteins include an *N*-terminal fusion tag of approximately 29 kDa containing the Trx-tag followed by a 6X His-tag. For long-term storage of the purified enzyme preparations at −80 °C, glycerol was added to the purified proteins to a final concentration of 10% and then snap-frozen in liquid nitrogen. Protein concentrations were determined using the Qubit v2.0 Fluorimeter and the Qubit Protein Assay Kit (Thermo Fisher Scientific Inc., Waltham, MA, USA. Enzyme kinetic analyses were performed on stored aliquots from the same batch of purified protein. These aliquots were thawed on ice before use. 

### 3.6. Bi-Substrate (Benzoyl–CoA and Glycine) Kinetic Analysis

To determine the bi-substrate kinetic parameters for each of the three GLYAT haplotype variants, enzyme assays were performed in which both substrate concentrations were varied simultaneously. The glycine concentrations were varied from 1–200 mM and the benzoyl–CoA concentrations from 20–200 μM in various combinations according to a grid. Enzyme activity was determined using a colorimetric assay that measures glycine-dependent release of CoA at 412 nm, in the presence of 5,5′-dithiobis(2-nitrobenzoic acid) (DTNB) [79]. Enzyme assays were 200 µL in volume and contained 25 mM Tris-acetate, pH 8.0, 100 µM DTNB, 2 µg of a particular recombinant GLYAT variant, glycine and benzoyl–CoA. Reactions were carried out at 37 °C in 96-well plates and monitored for 20-min at 40 s intervals using a BioTek plate reader and accompanying Gen5 software (BioTek, Winooski, VT, USA). Raw initial rate data for each of the combinations of substrate concentrations are provided in Appendix A. Activities from triplicate assays were calculated by linear regression of the A_412_ versus time data over the linear range of the time-course and expressed as µmol/min/mg protein.

The data were processed using the Python programming language, making use of Jupyter notebooks [80] (https://jupyter.org, accessed on 14 March 2021) and the numpy [81], scipy (https://www.scipy.org/, accessed on 14 March 2021), and pandas [82] libraries. Raw and processed data were visualised with the matplotlib plotting library [83]. Kinetic parameters were determined by non-linear regression, using the Python lmfit module [84] of the initial rate vs. substrate concentration data to the bi-substrate Hill equation:(1)$$v=\frac{k_{cat}\cdot e_{T}\cdot {\left(\frac{G}{s_{0.5,g}}\right)}^{h_{g}}{\left(\frac{B}{s_{0.5,b}}\right)}^{h_{b}}}{\left(1+{\left(\frac{G}{s_{0.5,g}}\right)}^{h_{g}}\right)\left(1+{\left(\frac{B}{s_{0.5,b}}\right)}^{h_{b}}\right)}$$
where *k*_cat_ is the catalytic rate constant (turnover number), *e*_T_ is the total enzyme concentration, *G* and *B* are the concentrations of glycine and benzoyl–CoA, respectively, *s*_0.5,*g*_ and *s*_0.5,*b*_ are the respective half-saturation constants, and *h_g_* and *h_b_* the respective Hill coefficients for glycine and benzoyl–CoA. Kinetic parameters were estimated from a global fit to all the data (with both substrates varied) simultaneously. Because excessive correlations were observed between the *k*_cat_ and other parameters, the fit was performed in two stages: initially, the *k*_cat_ was estimated from a fit of the uni-substrate Hill equation to the rate vs. glycine concentration data collected at the highest benzoyl–CoA concentration (200 µM). This value was subsequently fixed and the remaining parameters were estimated from a global fit of the bi-substrate Hill equation to all the data for both varied substrates.

## 4. Conclusions

Impaired phase II detoxification has been associated with adverse reactions to pharmaceutical drugs and may be involved in the pathogenesis of complex multifactorial diseases such as cancer [85,86]. In the case of glycine conjugation, it has been shown that GLYAT expression is transcriptionally down-regulated in hepatocellular carcinoma specimens [28]. Very little is still understood about the physiological implications of the impairment of glycine conjugation and the consequences of substrate interaction and competition in this pathway. This is especially relevant because the exposure of humans to benzoate is increasing due to the wide use of benzoate as a preservative [18]. This is reflected in the high levels of hippurate (up to 932.66 µmol/mmol creatinine) found in urine [87,88]. 

Glycine conjugation can be influenced by several factors, including the availability of ATP, CoA, and glycine, variants in the *ACSM2B* and *GLYAT* genes, and differential expression of ACSM2B and GLYAT. The present study and previous studies have shown that both the *ACSM2* [2] and the *GLYAT* genes [27] are highly conserved and that alleles, predicted to have a deleterious effect on enzyme function, are found at very low frequencies. The glycine conjugation pathway might be essential for life as a metabolic defect related to this pathway has yet to be identified. This hypothesis, however, needs to be tested by establishing a model system in which variants of ACSM2B and GLYAT can be co-expressed to investigate the effect of genetic variation on the detoxification ability and/or toxicity of the pathway. This will provide a better understanding of what the effect of genetic variants is on the in vivo glycine conjugation ability. Development of a model system in which the detoxification ability of the glycine conjugation pathway can be evaluated by measuring both the initial conjugation with CoA and the acyl-transfer to the amino acid will contribute to filling one of the biggest gaps in the literature, whereby previous studies characterised glycine conjugation as a one-step process. This is especially relevant for studies on the pharmacokinetic evaluation of salicylate metabolism. 

If it could be established that the glycine conjugation pathway is overloaded by overconsumption of benzoate and salicylate, this could lead to recommendations for healthier choices when consuming food, in addition to a decreased use of benzoate as a food additive. It will then also be possible to provide genetic testing to identify individuals with variants affecting their glycine conjugation ability and to advise individuals on whether they might potentially be at risk from these dietary additives. This is particularly relevant to isovaleric acidaemia patients. 

## Figures and Tables

**Figure 1 ijms-22-03129-f001:**
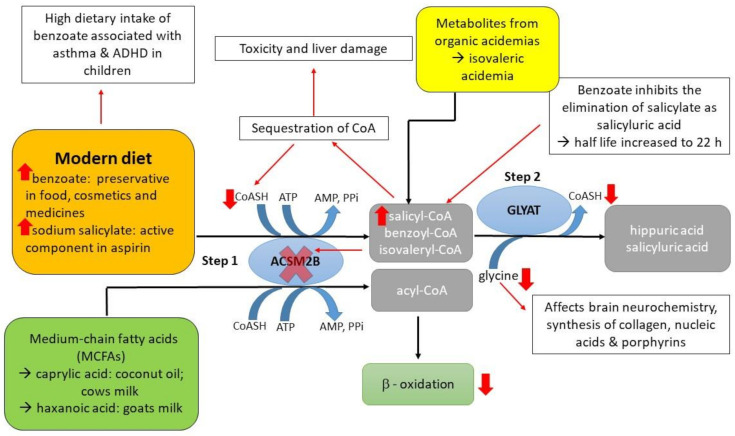
A schematic representation depicting the effect of a high dietary intake of benzoate on the glycine conjugation pathway. Benzoate and salicylate are activated to an acyl-CoA by the mitochondrial xenobiotic/medium-chain fatty acid: coenzyme A (CoA) ligase (ACSM2B) and subsequently conjugated to glycine by glycine *N*-acyltransferase (GLYAT). A high dietary intake of benzoate can lead to a decrease in available glycine. The acyl-CoAs can no longer be conjugated to glycine by GLYAT resulting in an increase of the acyl-CoA intermediates and sequestration of free CoA. The acyl-CoA intermediates can inhibit ACSM2B leading to a reduction in mitochondrial energy production. Metabolites from organic acidaemia, for example isovaleryl-CoA from isovaleric acidaemia, provide an additional detoxification load to the glycine conjugation pathway.

**Figure 2 ijms-22-03129-f002:**
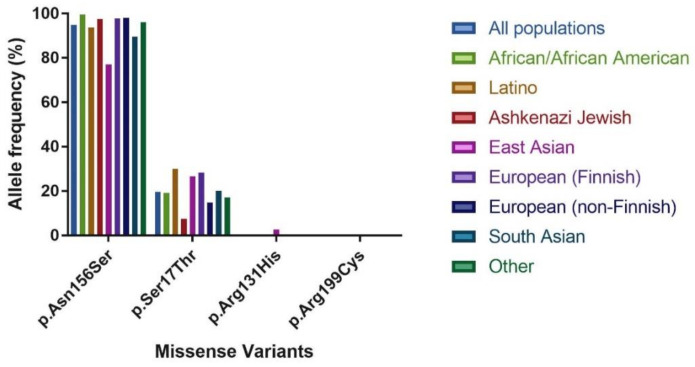
Comparison of the allele frequency between different populations of missense variants of interest to this study.

**Figure 3 ijms-22-03129-f003:**
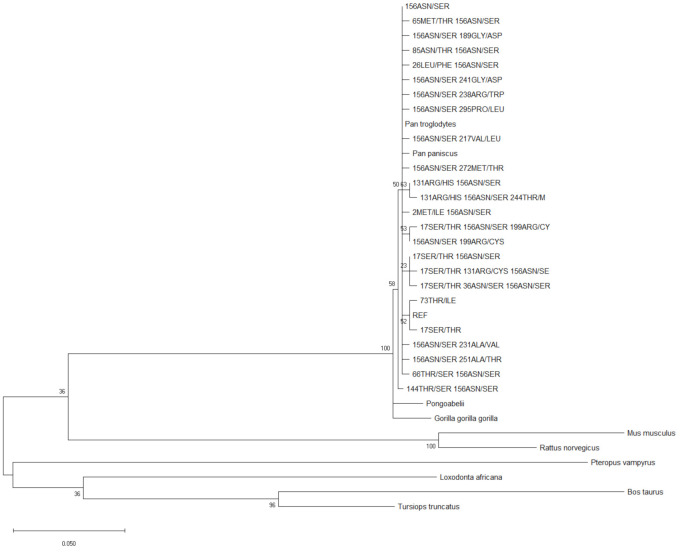
Molecular Phylogenetic analysis of the GLYAT haplotypes using the maximum likelihood method. The evolutionary history was inferred by using the maximum likelihood method based on the Jones–Thornton–Taylor (JTT) matrix-based model [57]. The bootstrap consensus tree inferred from 500 replicates [59] is taken to represent the evolutionary history of the taxa analysed. Branches corresponding to partitions reproduced in less than 50% bootstrap replicates are collapsed. The percentage of replicate trees in which the associated taxa clustered together in the bootstrap test (500 replicates) is shown next to the branches. Initial tree(s) for the heuristic search were obtained automatically by applying Neighbour-Join (NJ) and Bio NJ algorithms to a matrix of pairwise distances estimated using a JTT model, and then selecting the topology with superior log likelihood value. A discrete Gamma distribution was used to model evolutionary rate differences among sites (five categories; +*G*, parameter = 2.7686). The analysis involved 34 amino acid sequences. All positions containing gaps and missing data were eliminated. There were a total of 296 positions in the final dataset. Evolutionary analyses were conducted in MEGA X [52].

**Figure 4 ijms-22-03129-f004:**
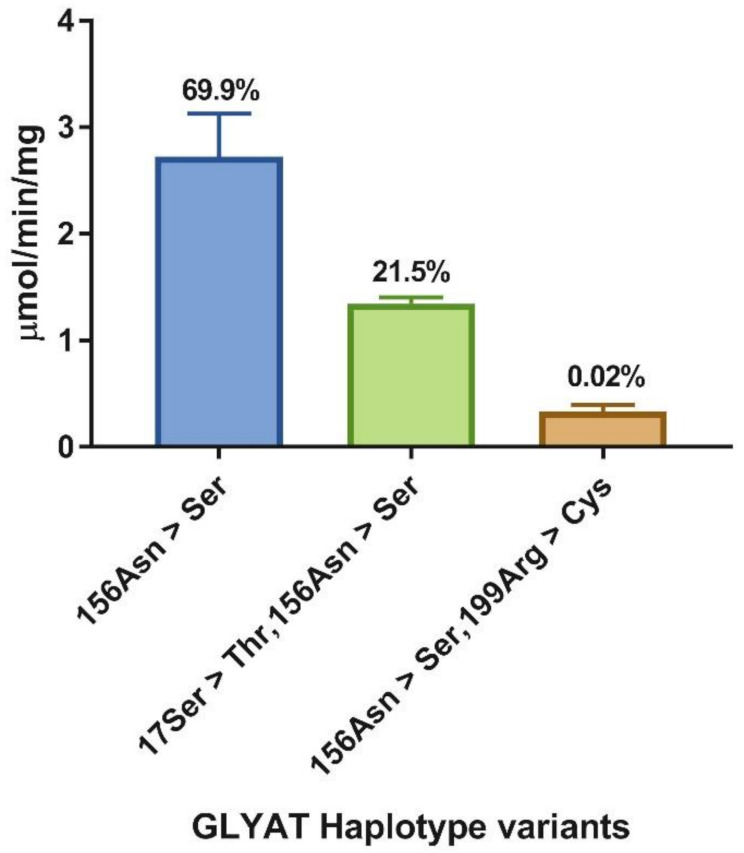
Relative enzyme activity of glycine *N*-acyltransferase (GLYAT) haplotypes 156Asn > Ser, 17Ser > Thr,156Asn > Ser and 156Asn > Ser,199Arg > Cys. Assays were performed in triplicate with 2 µg protein, 20 mM glycine, and 80 µM benzoyl–CoA. The standard deviation is shown by the error bars and the haplotype frequencies are indicated above each bar.

**Figure 5 ijms-22-03129-f005:**
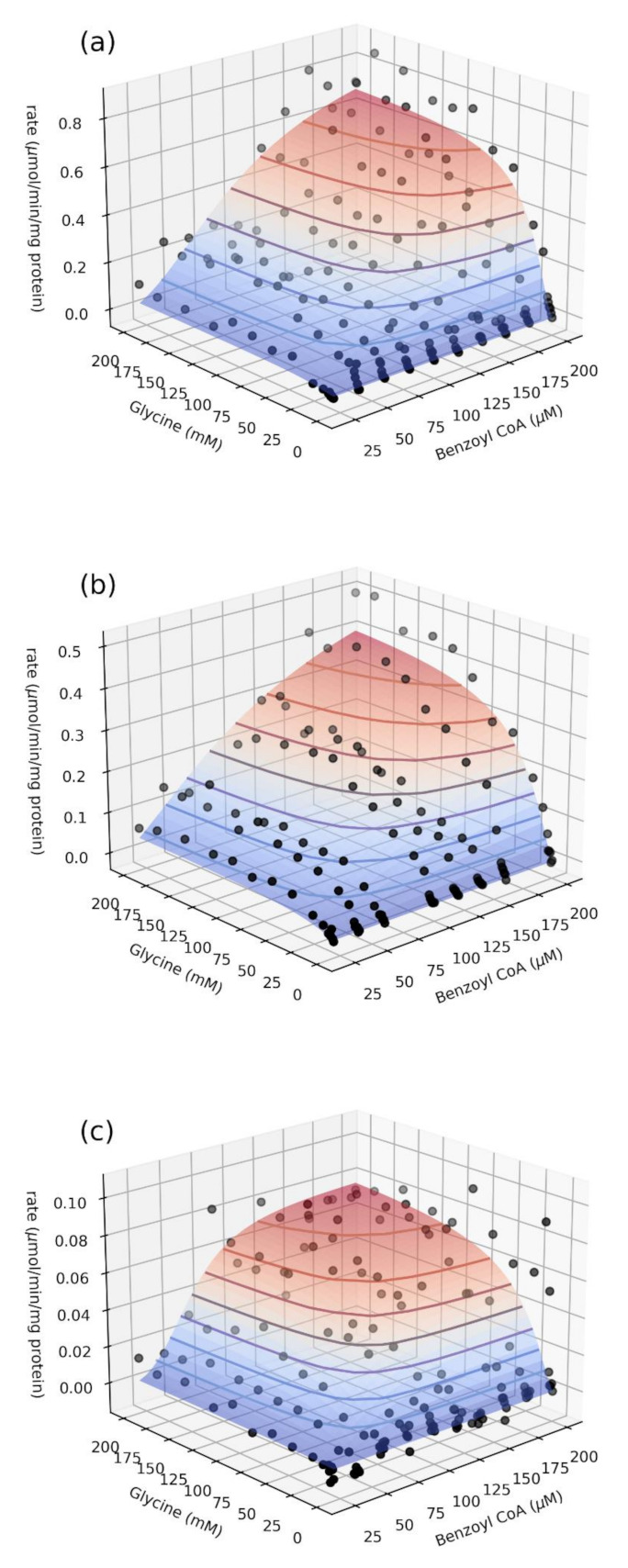
Global fit of initial rate data to the two-substrate Hill equation (see Section 3) for the haplotype variants 156Asn > Ser (**a**), 17Ser > Thr,156Asn > Ser (**b**), and 156Asn > Ser,199Arg > Cys (**c**).

**Table 1 ijms-22-03129-t001:** Population data used in this study.

Population	Exomes	Genomes	Total
African/African American	8 128	4 359	12,487
Latino	17,296	424	17,720
Ashkenazi Jewish	5 040	145	5 185
East Asian	9 197	780	9 977
Finnish	10,824	1 738	12,562
Non-Finnish European	56,885	7 718	64,603
South Asian	15,308	#	15,308
Other^*^	3 070	544	3 614
Female	57,787	6 967	64,754
Male	67,961	8 741	76,702
Total	125,748	15,708	138,632

# 31 South Asian samples were grouped with Other. * Individuals were classified as “other” if they did not unambiguously cluster with the major populations (that is i.e., African, African American, Latino, Ashkenazi Jewish, East Asian, Finnish, Non-Finnish European, South Asian) in a principal component analysis (PCA). (This is an extract of the population data available on the gnomAD browser).

**Table 2 ijms-22-03129-t002:** Results from Tajima’s neutrality test.

Gene	Number of Sequences (m)	Number of Segregating Sites (S)	Nucleotide Diversity (π)	Tajima Test Statistic (D)
GLYAT	25	21	0.007748	−2.13

**Table 3 ijms-22-03129-t003:** Enzyme–kinetic parameters of the GLYAT variants.

Haplotype	*V*_f_ µmol min^−1^ mg protein^−1^	*k*_cat_ s^−1^	*s*_0.5,gly_ mM	*h* _gly_	*s*_0.5,benz_ µM	*h* _benz_
156Asn > Ser	0.85 ± 0.06	0.48 ± 0.03	23 ± 2	1.6 ± 0.1	97 ± 3	2.1 ± 0.1
17Ser > Thr,156Asn > Ser	0.62 ± 0.02	0.35 ± 0.01	29 ± 3	1.3 ± 0.1	118 ± 7	1.5 ± 0.1
156Asn > Ser,199Arg > Cys	0.083 ± 0.005	0.047 ± 0.003	30 ± 3	1.4 ± 0.1	61 ± 3	3.5 ± 0.5

**Table 4 ijms-22-03129-t004:** Kinetic parameters of benzoyl–CoA and glycine for GLYAT as reported in the literature.

Parameters	Values	Recombinant Variant/Isolated from Liver	Reference
*K*_Mapp_ (benzoyl-CoA) (µM)	13	Purified GLYAT from human liver	[33]
	28 ± 5	Purified recombinant 17Ser > Thr variant	[31]
	38 ± 4	Purified recombinant 156Asn > Ser variant	[31]
*s* _0.5_	61 ± 3	Purified recombinant 156Asn > Ser,199Arg > Cys variant	This study (bi-substrate Hill)
*K*_Mapp_ (benzoyl-CoA) (µM)	67 ± 5	Partially purified GLYAT from human liver	[32]
	79 ± 38	Purified recombinant wildtype	[34]
	88 ± 66	Purified recombinant 156Asn > Ser variant	[34]
*s* _0.5_	97 ± 3	Purified recombinant 156Asn > Ser variant	This study (bi-substrate Hill)
	118 ± 7	Purified recombinant 17Ser > Thr,156Asn > Ser variant	This study (bi-substrate Hill)
*K*_Mapp_ (benzoyl-CoA) (µM)	139 ± 85	Purified recombinant L_61_ variant	[34]
	209	Purified recombinant 156Asn > Ser variant	[28]
	57900 *	Purified GLYAT from human liver	[30]
*K*_Mapp_ (glycine) (mM)	6.4	Purified GLYAT from human liver	[33]
	6.5 ± 1	Partially purified GLYAT from human liver	[32]
*s* _0.5_	23 ± 2	Purified recombinant 156Asn > Ser variant	This study (bi-substrate Hill)
*K*_Mapp_ (glycine) (mM)	26.6	Purified recombinant 156Asn > Ser variant	[28]
*s* _0.5_	29 ± 3	Purified recombinant 17Ser > Thr,156Asn > Ser variant	This study (bi-substrate Hill)
	30 ± 3	Purified recombinant 156Asn > Ser199Arg > Cys variant	This study (bi-substrate Hill)
*V*_max_ (nmol/min/mg)	83 ± 5	Purified recombinant 156Asn > Ser199Arg > Cys variant	This study (bi-substrate Hill)
	543 ± 21	Purified GLYAT from human liver	[33]
	620 ± 20	Purified recombinant 17Ser > Thr,156Asn > Ser variant	This study (bi-substrate Hill)
	665 ± 40	Purified recombinant 17Ser > Thr variant	[31]
	807	Recombinant 156Asn > Ser variant	[28]
	850 ± 60	Purified recombinant 156Asn > Ser variant	This study (bi-substrate Hill)
	1230	Purified recombinant 156Asn > Ser variant	[31]
	17,100 ^#^	Purified GLYAT from human liver	[30]
	121,000 ^+^	Purified recombinant L_61_ variant	[34]
	490,000 ^+^	Purified recombinant wildtype	[34]
	1,359,000 ^+^	Purified recombinant 156Asn > Ser variant	[34]

* The unit reported in the article is mM; therefore, this value was converted to µM. We do think that this is either a typing or a calculation error and should be 57.9 µM but because the enzyme assay conditions are not described in the article it is difficult to ascertain where the error lies. ^#^ This error also affected the V_max_ value. ^+^ These V_max_ values are too high to be correct and might also be a calculation error.

## Data Availability

The data presented in this study are available in Appendix A, and Appendix A.

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
