# Peer review of "Functional Characterisation of Three Glycine N-Acyltransferase Variants and the Effect on Glycine Conjugation to Benzoyl–CoA"

_ijms, 2021, doi:10.3390/ijms22063129_

Round 1

Reviewer 1 Report

The manuscript by Rohwer et al. contains a very nice study on the prevalence of haplotypes of GLYAT in humans, as well as a biochemical characterization of three interesting variants, which include the most frequent ones and a highly deleterious one. The study is well done and I only have a few minor comments which don’t affect the scientific merit:

Title: the enzyme should be spelled out at least here

Line

18: not the gene, but the protein was actually analysed, using haplotype variants that lead to a changed aa sequence

24: T17S156 variant (as well as other places; just using the numbers is lab jargon). I would also recommend to mention already in the abstract that S156 is by far the most frequent haplotype and was therefore used as reference [rather than the poorly mentioned REF protein, which has not been analysed at all].

33: the organism (human) should at least be mentioned once. This journal is not one of those where readers expect to read nothing else.

38: why ligases when there is only one of them mentioned later on (also check for correctness of singular/plural use throughout the paper)? This would be a good place to introduce the ACSM2A and B that appear later without explanantion.

43: chage variants to variations

55: isoval… adidemia

62 delete “for e.g.”

63/171 has been

73 hippuric acid as product of GLYAT has not been introduced

120 shift “purified” before “S156”

231: REF has not been introduced. Since it is apparently supposed to be the refence sequence of GLYAT, this version would normally be required to include into the study as reference. However, since the authors found that it only comprises 7% of the haplotypes, I find it justified to use the most frequent S156 variant as reference for the study. However, the authors need to point out this fact (abundances) earlier and more clearly and add some statement to justify this decision. For clarity “REF” should also be assigned by some different name.

260: Fig. 3 contains Pan troglodytes, not paniscus! Moreover, if ape sequences are already added, why not add also the ones from the genome-sequenced bonobo and orang-utan species? Also, species name are in lowercase.

284: change were to where

290: I don’t understand this sentence because of garble grammar

290(and elsewhere): I would recommend not to use “SNP” in this paper, but to replace it by “mutation”. SNP implies a (polymorphic) change on nucleotide level, while we have here defined mutations changing the aa sequence.

293: for T17S156 it is not obvious why it should have deleterious effects if T17 is similar to WT.

311: explain GNAT

326: why point out benzoyl-CoA, if both substrates behaved cooperatively?

335: the surface plots do not actually allow to “clearly observe” cooperativity. Therefore, it is good to have the 2D-plots in the supplement, but maybe it is possible to enhance the figure by adding contour lines into the surface areas.

407 change that to the

421 errors

467 were used

516 Supplementary table 3 is not really necessary to show

554/558 confusion with ACSM2A/B, see comment above

Reviewer 2 Report

This is a fine manuscript, a solid body of work.  This reviewer only has a few comments for the authors.

  1. The number of significant figures in Table 3 should be 2 significant figures not 3.  Given the scatter observed in the v vs. [S] (seen in the supplementary data), the authors might consider using 1 significant figure.
  2. In the supplementary figures, the authors should consider including the a few 1/v vs. 1/[S] plots.  A general reader of the manuscript might better recognize kinetic data with a Hill number greater than 1.0 in those sorts of plots.  The kinetic data shown in Fig. 5a-c will, likely, not be familiar to the general reader of the manuscript.
  3. The kinetic results show cooperativity, but provide no explanation for these results.  In the discussion, the authors should provide some explanation for cooperativity in their system.
